# A third vaccination with a single T cell epitope confers protection in a murine model of SARS-CoV-2 infection

Iris N. Pardieck [1], Tetje C. van der Sluis [1,9], Esmé T. I. van der Gracht [1,9], Dominique M. B. Veerkamp [1], Felix M. Behr [1], Suzanne van Duikeren [1], Guillaume Beyrend [1], Jasper Rip [1], Reza Nadafi[1], Elham Beyranvand Nejad [1], Nils Mülling [1], Dena J. Brasem[1], Marcel G. M. Camps[1], Sebenzile K. Myeni [2], Peter J. Bredenbeek[2], Marjolein Kikkert [2], Yeonsu Kim[3], Luka Cicin-Sain [3], Tamim Abdelaal [4,5], Klaas P. J. M. van Gisbergen [6], Kees L. M. C. Franken[1], Jan Wouter Drijfhout [1], Cornelis J. M. Melief[7], Gerben C. M. Zondag[8], Ferry Ossendorp[1] & Ramon Arens [1✉]

Understanding the mechanisms and impact of booster vaccinations are essential in the design and delivery of vaccination programs. Here we show that a three dose regimen of a synthetic peptide vaccine elicits an accruing CD8[+] T cell response against one SARS-CoV-2 Spike epitope. We see protection against lethal SARS-CoV-2 infection in the K18-hACE2 transgenic mouse model in the absence of neutralizing antibodies, but two dose approaches are insufficient to confer protection. The third vaccine dose of the single T cell epitope peptide results in superior generation of effector-memory T cells and tissue-resident memory T cells, and these tertiary vaccine-specific CD8[+] T cells are characterized by enhanced polyfunctional cytokine production. Moreover, fate mapping shows that a substantial fraction of the tertiary CD8[+] effector-memory T cells develop from re-migrated tissue-resident memory T cells. Thus, repeated booster vaccinations quantitatively and qualitatively improve the CD8[+] T cell response leading to protection against otherwise lethal SARS-CoV-2 infection.

[1] Department of Immunology, Leiden University Medical Center, Leiden, The Netherlands. [2] Department of Medical Microbiology, Leiden University Medical Center, Leiden, The Netherlands. [3] Department of Viral Immunology, Helmholtz Centre for Infection Research, Braunschweig, Germany. [4] Delft Bioinformatics Lab, Delft University of Technology, Delft, The Netherlands. [5] Department of Radiology, Leiden University Medical Center, Leiden, The Netherlands. [6] Department of Hematopoiesis, Sanquin Research and Landsteiner Laboratory, Amsterdam, The Netherlands. [7] ISA Pharmaceuticals BV, Leiden, The Netherlands. [8] Immunetune BV, Leiden, The Netherlands. [9]These authors contributed equally: Tetje C. van der Sluis, Esmé T. I. van der Gracht. ✉email: R.Arens@lumc.nl

The coronavirus disease 2019 (COVID-19) pandemic, caused by the β-coronavirus severe acute respiratory syndrome coronavirus 2 (SARS-CoV-2), remains a global health emergency. Vaccines eliciting neutralizing antibodies against the Spike protein of SARS-CoV-2 have shown high effectiveness[1,2]. However, the level of neutralizing antibodies declines after vaccination[3] and at-risk patient groups, including transplant recipients, individuals suffering from B cell leukemia, and auto-immune disease patients on selected immunosuppressive regimens (such as B cell depleting reagents), exhibit lower humoral and cellular immunity after vaccination[4–6]. Moreover, the mutation rate of coronaviruses is substantial and certain mutations in the Spike protein of SARS-CoV-2 have emerged that can lead to a decline in neutralizing antibody-mediated protection[7,8]. In order to address this clinical problem, a third vaccination for solid organ transplant recipients is becoming the standard of care in more and more countries, and supplementary T cell-focused strategies have been suggested[9,10].

While boosters with the current vaccines likely lead to the generation of enhanced levels of neutralizing antibodies, it is uncertain if this will improve protection in the aforementioned risk groups and provide protection against the emergence of novel SARS-related viruses. However, booster vaccination may also lead to enhanced T cell responses, which is expected to contribute to the control of SARS-CoV-2 infection as evidenced by their association to disease severity and protection in humans[11–16] and animal models[17,18]. Moreover, T cell-focused vaccines can be directed to more conserved regions of the coronavirus, which may lead to a broad T cell-based cross-protection against multiple coronavirus strains and likely independently expands the protection provided by antibodies. Such T cell-mediated protection is important given the potential emergence of similar viruses in the future that pose a significant threat to global public health. Previous studies showed that T cells can mediate protection by themselves against SARS-CoV-1[19,20] but whether T cell-eliciting vaccines can protect against SARS-CoV-2 is unclear. Here, we demonstrate that a peptide vaccine exclusively eliciting CD8+ T cell responses against a single epitope present in the Spike protein delivers full protection against SARS-CoV-2, provided that the vaccine was administered in a double booster regimen, leading to the induction of high numbers of circulating and tissue-resident memory T ($T_{RM}$) cells. A third vaccination was especially critical to achieve cytokine polyfunctionality and induction of elevated $T_{RM}$ cell numbers in the lungs and liver, and additionally fueled retrograde migration of $T_{RM}$ cells into the circulation. Thus, booster vaccinations eliciting strong CD8+ T cell responses are a promising strategy against coronavirus-mediated disease independent of neutralizing antibodies.

## Results

**Vaccines eliciting antibodies against confirmational proteins but not linear B cell epitopes are neutralizing and effective against SARS-CoV-2 infection.** Single B and T cell epitope vaccines can provide effective approaches against a variety of pathogens and malignancies[21–24]. To investigate the efficacy of linear B cell epitope vaccines against SARS-CoV-2 infection, we selected five different linear epitopes, present at different locations in the Spike protein, which were considered to potentially able to elicit protective antibodies[25–27]. To test their immunogenicity, C57BL/6 mice were vaccinated with synthetic long peptide (SLP)-based vaccines each containing a single B cell epitope. After three vaccinations, however, these SLP vaccines did not elicit Spike-specific antibody responses (Supplementary Fig. 1a). To improve

this synthetic vaccine, we tested the particular influence of adjuvants and of CD4 T cell help for the specific antibody induction of the B cell-SLP vaccines. To this end, the SLP-based vaccines were coupled with the universal helper epitope PADRE and adjuvanted with CpG and Incomplete Freund's Adjuvant (IFA). The combination of these adjuvants acted synergistically to elicit high levels of antibodies against the Spike protein (Fig. 1a). This synergistic effect correlated with a higher induction of PADRE-specific CD4+ T cell responses (Fig. 1b, c). Coupling of the PADRE epitope to the linear B cell epitopes, both N-terminal and C-terminal, also contributed to the improvement of the Spike-specific IgG antibody response, instead of combining the linear B cell epitopes and PADRE epitope in a single vaccine (Fig. 1d and Supplementary Fig. 1b). Moreover, depletion of CD4+ T cells during the vaccination period resulted in a decreased Spike-specific antibody response (Supplementary Fig. 1c), highlighting the need from CD4+ T cells for an optimal IgG response. CD8+ T cell responses were not induced by the B cell-SLP vaccines (Supplementary Fig. 1d).

To determine whether immunization with optimized linear B cell epitope vaccines enables protective immune responses against SARS-CoV-2, we used transgenic mice expressing the human angiotensin I-converting enzyme 2 (ACE2) receptor under the regulation of the cytokeratin 18 (K18) promoter, which develop severe lung disease in response to SARS-CoV-2 infection[28]. The K18-hACE2 mice were vaccinated three times with a two week interval with either one CpG/IFA adjuvanted PADRE-coupled B cell-SLP vaccine or a combination of the five different B cell-SLP vaccines. Spike-specific IgG antibody responses were elicited and reached high titers after the third (3rd) vaccination (Fig. 1e). Four weeks after the 3rd vaccination, the K18-hACE2 mice were challenged intranasally with a lethal dose of SARS-CoV-2 and body weight and clinical symptoms were measured daily as parameters of disease. Both unvaccinated mice and mice that were vaccinated three times with the B cell-SLPs succumbed to SARS-CoV-2 infection as evidenced by >20% reduction in bodyweight and a moribund condition (Fig. 1f, g), showing that SLP vaccination with these linear B cell epitopes was not able to protect against SARS-CoV-2 infection. This inability of protection correlated with the incapability of antibodies to neutralize virus cell entry (Fig. 1h). In contrast, a DNA vaccine encoding the entire Spike protein fully protected K18-hACE2 transgenic mice from SARS-CoV-2 infection, and this correlated with the ability to elicit neutralizing Spike-specific antibodies (Fig. 1i–l). In addition, this DNA vaccine also induced a Spike-specific CD4+ and CD8+ T cell response (Fig. 1m and Supplementary Fig. 1e). The response against an immunodominant CD8+ T cell epitope present in the Spike protein of SARS-CoV-1 and SARS-CoV-2 (i.e., Spike 539-546, VNFNFNGL)[29] increased upon the second (2nd) but not the 3rd vaccination, and was characterized by an increased expression of the activation-associated NK cell receptor KLRG1 on the Spike$_{539-546}$-specific CD8+ T cells (Fig. 1m, n). Overall, these results indicate that antibody responses to these single linear B cell epitopes are inferior for the induction of neutralizing antibodies, although we cannot exclude the possibility that single linear B cell epitopes exist that can elicit neutralizing antibodies. Nevertheless, antibody responses induced by vaccine platforms encoding properly folded proteins presenting conformational B cell epitopes are able to generate neutralizing antibodies and provide protection against SARS-CoV-2 infection.

**A third vaccination with a peptide harboring a single T cell epitope protects against SARS-CoV-2 challenge.** To determine whether an immune response induced by a single CD8+ T cell

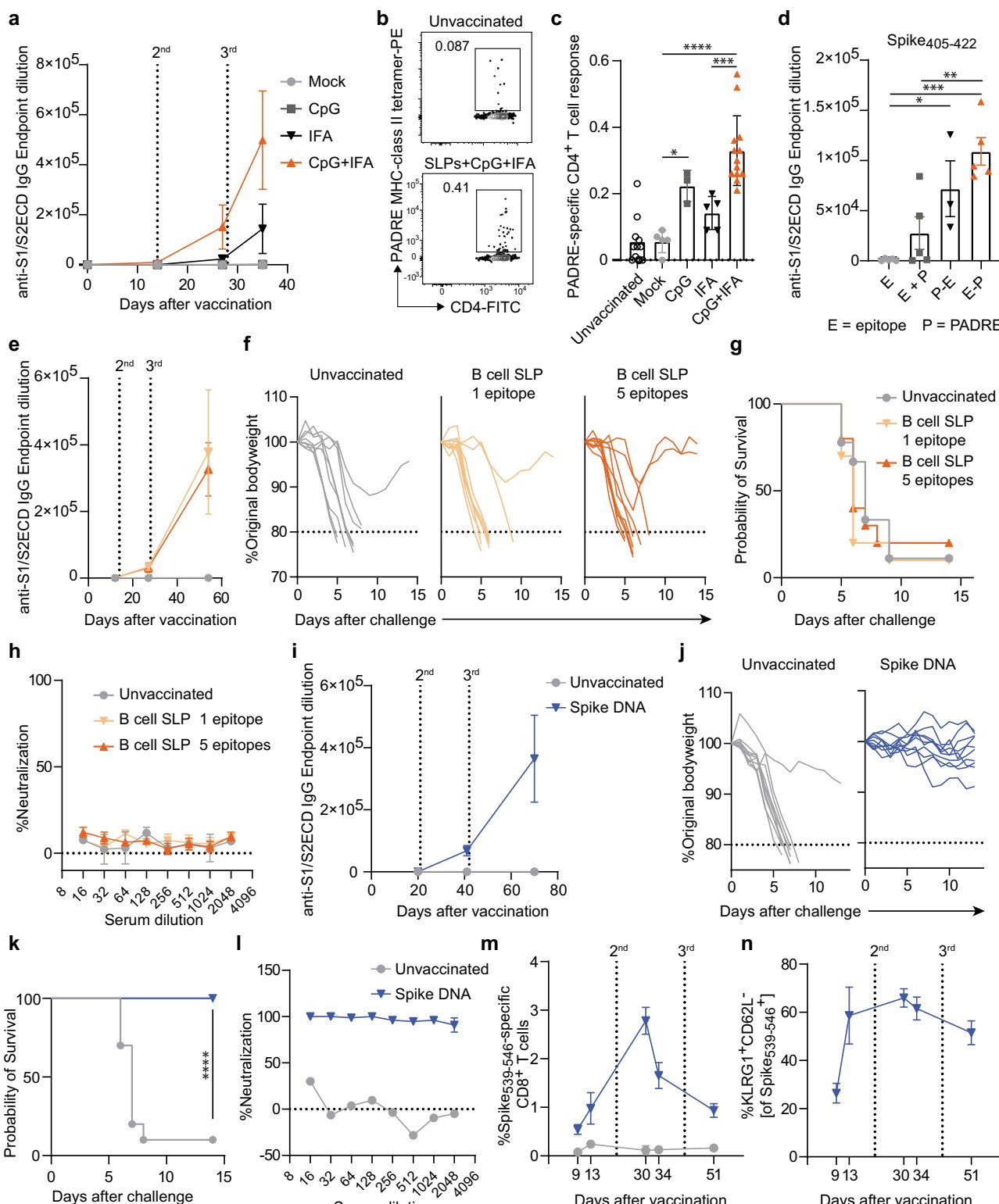

epitope vaccine was able to protect against SARS-CoV-2 infection, K18-hACE2 transgenic mice were vaccinated once, twice or three times with an SLP encoding the immunodominant $CD8^+$ T cell epitope $Spike_{539-546}$. Five weeks after the final vaccination with the $Spike_{539-546}$-SLP, mice were intranasally challenged with a lethal dose of SARS-CoV-2 (Fig. 2a). SLP vaccination induced a $Spike_{539-546}$-specific $CD8^+$ T cell response in all mice, which increased with each booster immunization and remained elevated after contraction (Fig. 2b–d). $CD4^+$ T

cell responses and Spike-specific antibodies were not induced (Supplementary Fig. 1f, g). Whereas 75% of the non-vaccinated mice succumbed to SARS-CoV-2 challenge, 66% and 50% of the mice that received one or two vaccinations, respectively, succumbed. Strikingly, all mice that received a $3^{rd}$ vaccine dose recovered from the SARS-CoV-2 challenge despite initial weight loss (Fig. 2e, f). Thus, a third vaccination with an SLP containing a single $CD8^+$ T cell epitope protects against lethal SARS-CoV-2 infection.

**Fig. 1 Vaccines eliciting antibodies against confirmational proteins but not linear B cell epitopes are neutralizing and effective against SARS-CoV-2 infection. a** C57BL/6 mice were vaccinated subcutaneously (s.c.) on day 0, 14, and 28 with synthetic long peptides (SLPs) consisting of PADRE-coupled linear B cell epitopes adjuvanted or not with CpG and/or Inactivated Freund's Adjuvant (IFA) (Mock, IFA: $n = 5$; CpG, CpG+IFA: $n = 3$). Spike-specific IgG kinetics in blood are shown. **b** Representative flow cytometry plots of PADRE-specific CD4$^+$ T cells in naïve unvaccinated mice and in day 21 post-vaccinated mice. **c** PADRE-specific CD4$^+$ T cell response on day 21 after vaccination as described in (**a**) (Naïve: $n = 11$; Mock, IFA: $n = 5$; CpG: $n = 3$; CpG+IFA: $n = 13$). *$P = 0.048$, ***$P = 0.0007$, ****$P = <0.0001$. **d** C57BL/6 mice were vaccinated as described in (**a**) with SLPs consisting of the B cell epitope (E) alone, the epitope combined (uncoupled) with the PADRE (P) peptide (E+P) or N-terminal and C-terminal PADRE coupled to the B cell epitope (P−E, E−P), and all adjuvanted with CpG and IFA (E, E+P, E−P: $n = 5$; P−E: $n = 3$). Endpoint dilutions of Spike-specific IgG in blood at day 42 after vaccination with Spike$_{405-422}$-SLP are shown. *$P = 0.035$, **$P = 0.005$, ***$P = 0.0004$. **e–h** K18-hACE2 transgenic mice were vaccinated on day 0, 14, and 28 with one or five SLP vaccines containing a PADRE-coupled linear B cell epitope adjuvanted with CpG and IFA. Four weeks after the final vaccination, mice were intranasally infected with SARS-CoV-2 (**e–g**, Unvaccinated: $n = 9$; B cell SLP 1 epitope, B cell SLP 5 epitopes: $n = 10$). **e** Spike-specific IgG kinetics in blood. **f** Weight loss kinetics after SARS-CoV-2 challenge. **g** Survival graph after SARS-CoV-2 challenge. **h** SARS-CoV-2 neutralizing capacity by antibodies after B cell-SLP vaccination on day 42 measured by a WT virus neutralization assay (VNA) (Unvaccinated, B cell SLP 1 epitope, $n = 2$; B cell SLP 5 epitopes, $n = 5$). **i–n** K18-hACE2 mice were vaccinated intradermally on day 0, 21, and 42 with a Spike-encoding DNA vaccine. **i** Spike-specific IgG (Unvaccinated: $n = 3$; Spike DNA: $n = 6$), **j** weight loss kinetics and **k** survival graph after SARS-CoV-2 challenge ($n = 10$ per group). ****$P = < 0.0001$. **l** SARS-CoV-2 neutralizing capacity by antibodies after Spike DNA vaccination on day 55 measured by a VNA. Data represented as mean ± SEM (Unvaccinated: $n = 1$; Spike DNA: $n = 6$). **m** Kinetics of Spike$_{539-546}$-specific CD8$^+$ T cells in blood. Data represented as mean ± SEM (Unvaccinated: $n = 3$; Spike DNA: $n = 6$). **n** Kinetics of KLRG1$^+$CD62L$^-$ expression on Spike$_{539-546}$-specific CD8$^+$ T cells in blood. Data represented as mean ± SEM ($n = 6$). One-way ANOVA with Tukey's multiple comparison test for (**c, d**); log-rank test for (**f, k**). Source data are provided as a Source Data file.

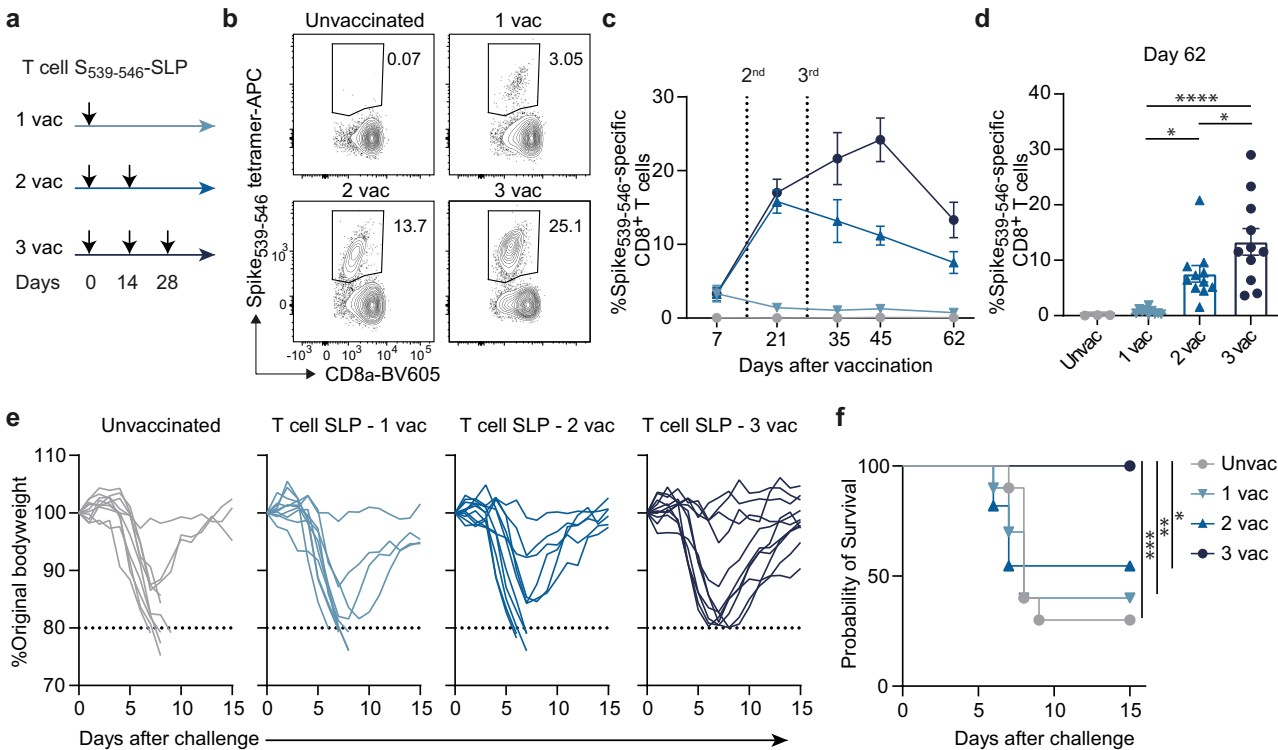

**Fig. 2 A third vaccination with a single T cell epitope protects against SARS-CoV-2 challenge. a** K18-hACE2 transgenic mice were vaccinated subcutaneously on day 0 (1$^{st}$ vaccination), day 14 (2$^{nd}$ vaccination), and day 28 (3$^{rd}$ vaccination) with the Spike$_{539-546}$-SLP vaccine adjuvanted with CpG. The Spike$_{539-546}$-specific CD8$^+$ T cell response was determined in blood in time. Five weeks after the final vaccination, vaccinated and unvaccinated mice were intranasally infected with 5000 PFU of SARS-CoV-2, and monitored for weight loss and signs of clinical discomfort. **b** Representative flow cytometry plots of Spike$_{539-546}$-specific CD8$^+$ T cells determined in the blood circulation at day 45 by MHC class I tetramer staining. **c** Spike$_{539-546}$-specific CD8$^+$ T cell kinetics in blood at indicated days after vaccination. Data is represented as mean ± SEM (Unvaccinated: $n = 3$; 1 vac: $n = 10$; 2 vac, 3 vac: $n = 11$). **d** Spike$_{539-546}$-specific CD8$^+$ T cells in blood at day 62 after vaccination. Data represented as mean ± SEM (Unvaccinated: $n = 3$; 1 vac: $n = 10$; 2 vac, 3 vac: $n = 11$). Symbols represent individual mice. *$P = 0.022-0.047$, ****$P = < 0.0001$. **e** Weight loss kinetics in time of SARS-CoV-2 challenged K18-hACE2 transgenic mice after T cell epitope SLP vaccination. **f** Survival graph of SARS-CoV-2 challenged K18-hACE2 transgenic mice after T cell epitope SLP vaccination (Unvaccinated, 1 vac: $n = 10$; 2 vac, 3 vac: $n = 11$). *$P = 0.0131$, $P = $**$0.0029$, ***$P = 0.0008$. One-way ANOVA with Tukey's multiple comparison test for (**d**); log-rank test for (**f**). Source data are provided as a Source Data file.

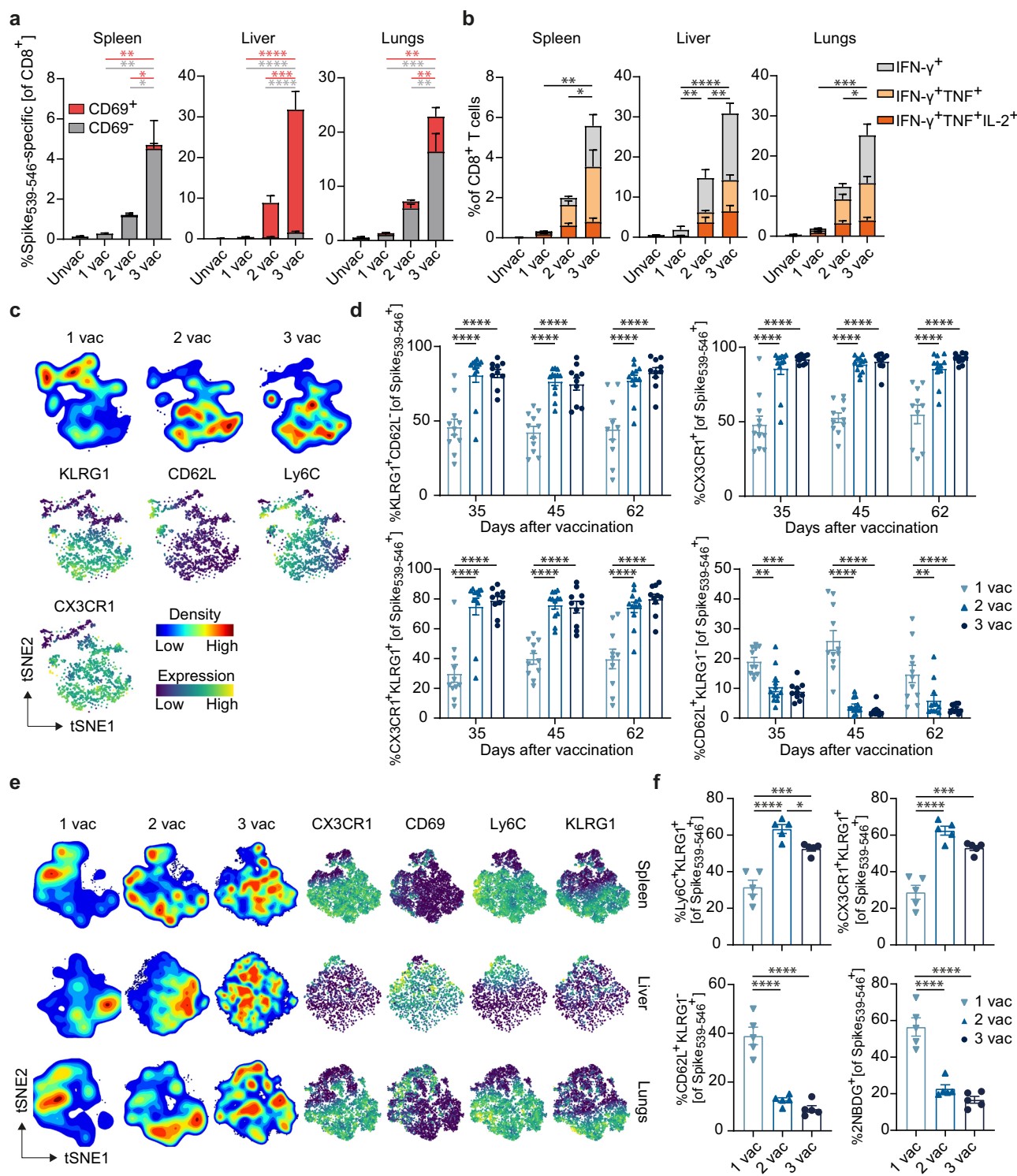

**A third vaccination with a single T cell epitope augments effector-memory and tissue-resident memory CD8$^+$ T cell formation**. Next, we aimed to determine the impact and underlying mechanisms of sequential SLP vaccinations on the memory differentiation of the elicited antigen-specific CD8$^+$ T cells. Three times vaccinated mice had 4-fold higher frequencies of circulating (CD69$^-$) Spike$_{539-546}$-specific CD8$^+$ T cells in the spleen compared to two times vaccinated mice, and 16-fold higher frequencies compared to once vaccinated mice (Fig. 3a). In the liver, the 3$^{rd}$ vaccination resulted in a notably higher Spike$_{539-546}$-specific CD8$^+$ T cell response compared to the single (120-fold

increase) or double vaccination (34-fold increase), and this difference was principally due to an increment of the CD69$^+$ T$_{RM}$ cells. In the lungs, the 3$^{rd}$ vaccination resulted in an 20-fold and 4-fold enhancement of Spike$_{539-546}$-specific CD69$^+$ T$_{RM}$ cells compared to single and double vaccination, respectively. Circulating (CD69$^-$) Spike$_{539-546}$-specific T cells in the lungs were 15-fold and 5-fold increased after the 3$^{rd}$ vaccination compared to the first (1$^{st}$) and 2$^{nd}$ vaccination, respectively (Fig. 3a). To investigate the effector potential of these Spike$_{539-546}$-specific CD8$^+$ T cells, the polyfunctional cytokine production capacity of the vaccine-specific CD8$^+$ T cells was determined. The increment

**Fig. 3 A third vaccination with a single T cell epitope augments effector-memory and tissue-resident memory CD8$^+$ T cell formation. a** Frequencies of CD69$^+$ and CD69$^-$ Spike$_{539-546}$-specific CD8$^+$ T cells in the CD8$^+$ T cell population in the spleen, liver and lungs at day 70 after 1, 2 or 3 SLP vaccinations. Data represented as mean + SEM ($n = 5$ per group). *$P = 0.0329$-0.0396, **$P = 0.0065$-0.0087, ***$P = 0.0003$, ****$P = < 0.0001$. **b** Intracellular cytokine production of CD8$^+$ T cells upon stimulation with the Spike$_{539-546}$ peptide epitope in different tissues on day 70 after 1, 2 or 3 SLP vaccinations. Data represented as mean + SEM ($n = 5$ per group). *$P = 0.0196$-0.0441, **$P = 0.0019$-0.0079, ***$P = 0.0004$, ****$P \leq 0.0001$. **c** tSNE maps describing the local probability density of Spike$_{539-546}$-specific CD8$^+$ T cells stained for KLRG1, CD62L, Ly6C and CX3CR1 at day 62 after 1, 2 and 3 vaccinations. **d** Frequencies of KLRG1$^+$CD62L$^-$, CX3CR1$^+$, CX3CR1$^+$KLRG1$^+$ and CD62L$^+$KLRG1$^-$ Spike$_{539-546}$-specific CD8$^+$ T cells in blood in time. Data represented as mean ± SEM (1 vac: $n = 11$; 2 vac: $n = 12$; 3 vac: $n = 10$). Symbols represent individual mice. **$P = 0.0021$-0.0031, ***$P = 0.0007$, ****$P \leq 0.0001$. **e** (Left) tSNE maps describing the local probability density of Spike$_{539-546}$-specific CD8$^+$ T cells in spleen, liver and lungs stained at day 69 after 1, 2 and 3 vaccinations with a flow cytometry panel for CD69, CD62L, CD44, Ly6C, KLRG1, and CX3CR1. (Right) marker expression for CX3CR1, CD69, Ly6C and KLRG1. **f** Cell surface marker (Ly6C$^+$KLRG1$^+$, CX3CR1$^+$KLRG1$^+$, and CD62L$^+$KLRG1$^-$) expression and glucose analog 2-(N-(7-nitrobenz-2-oxa-1,3-diazol-4-yl)amino)-2-deoxyglucose (2NBDG) uptake of Spike$_{539-546}$-specific CD8$^+$ T cells at day 69 in spleen after 1, 2 or 3 vaccinations. Data represented as mean ± SEM ($n = 5$ per group). *$P = 0.0402$, ***$P = 0.001$-0.004, ****$P \leq 0.0001$. One-way ANOVA with Tukey's multiple comparison test for (**a**, **b**, **d**, **f**). Source data are provided as a Source Data file.

in frequency of the Spike$_{539-546}$-specific CD8$^+$ T cells after the 2$^{nd}$ and 3$^{rd}$ vaccination coincided with an increase in polyfunctional CD8$^+$ T cells (IFN-γ$^+$TNF$^+$, IFN-γ$^+$TNF$^+$IL-2$^+$) in spleen, liver, and lungs (Fig. 3b). Thus, sequential booster vaccinations elicits significant and durable expansion of cytokine-producing polyfunctional CD8$^+$ T cells.

Next, we investigated the effect of booster vaccination on the differentiation status of the elicited Spike$_{539-546}$-CD8$^+$ T cells. Analysis of these cells in the blood circulation using t-distributed stochastic neighbor embedding (tSNE) algorithms highlighted that the memory phenotype of the vaccine-specific CD8$^+$ T cells in blood was skewed towards an effector-memory like state (CD62L$^-$KLRG1$^+$Ly6C$^+$CX3CR1$^+$) by one booster vaccination, and that the 2$^{nd}$ booster vaccination (3$^{rd}$ vaccination) enforced this phenotype slightly further (Fig. 3c, d). Moreover, in the tissues, booster vaccination also resulted in substantial alterations in the differentiation state and subset formation of the vaccine-elicited CD8$^+$ T cells. The Spike$_{539-546}$-specific CD8$^+$ T cells in the spleen mainly differentiated into CD62L$^-$KLRG1$^+$Ly6C$^+$CX3CR1$^+$ cells after the 1$^{st}$ booster (2$^{nd}$ vaccination) (Fig. 3e, f). These effector-memory-like cells were also present in the lungs, while in the liver, the large majority of the infiltrated cells were tissue-resident (CD69$^+$) and lacked expression of CX3CR1 and KLRG1 (Fig. 3e). Uptake of the glucose analog 2-(N-(7-nitrobenz-2-oxa-1,3-diazol-4-yl) amino)-2-deoxyglucose (2-NBDG) (Fig. 3f) was reduced following the 1$^{st}$ booster vaccination, indicating induction of a different metabolic efficiency after booster vaccination. Thus, while primarily the 2$^{nd}$ vaccination had a profound system-wide impact on the phenotype of the vaccine-induced CD8$^+$ T cells, the 3$^{rd}$ vaccination enhanced their numbers.

**Progressive differentiation of vaccine-specific CD8$^+$ T cells after booster vaccination**. To gain further insight into the differentiation of the vaccine-specific CD8$^+$ T cells upon sequential vaccination, we studied these cells in-depth using detailed single-cell high dimensional clustering analysis and trajectory inference with an antibody panel targeting multiple markers of cellular activation and differentiation. Visualization based on the cluster distribution of the splenic Spike$_{539-546}$-specific CD8$^+$ T cells indicated that the cells after the 1$^{st}$ vaccination were disparate from the 2$^{nd}$ and 3$^{rd}$ vaccination, while the 2$^{nd}$ and 3$^{rd}$ vaccination were largely overlapping (Fig. 4a). Strikingly, and consistent with the aforementioned data, a large KLRG1$^+$ cluster (cluster #3) co-expressing Ly6C and lacking CD62L, was more profoundly present in the 2$^{nd}$ and 3$^{rd}$ vaccination group (Fig. 4a–e). This cluster of cells also expressed CD44 and the cell proliferation marker Ki-67, but lacked expression of EOMES, TCF1, CXCR3, the activation-associated CD43$^{1B11}$ isoform, and

PD-1. Cluster #2, which had a similar expression except for KLRG1, was also increased in the 2$^{nd}$ and 3$^{rd}$ vaccination group compared to the 1$^{st}$ vaccination group (Fig. 4b–e). Other significantly different clusters, which were relatively reduced after the 2$^{nd}$ and 3$^{rd}$ vaccination compared to the 1$^{st}$ vaccination, were characterized by either high levels of CD62L and TCF1, and lacking markers such as KLRG1 and CD44 (cluster #1) or expressed intermediate differentiation phenotypes (cluster #7: CD44$^+$Ki-67$^+$Ly6C$^+$EOMES$^+$TCF1$^+$CXCR3$^+$PD-1$^+$KLRG1$^-$CD62L$^+$, cluster #6: CD44$^+$Ki-67$^+$Ly6C$^+$EOMES$^-$TCF1$^-$CXCR3$^{int}$PD-1$^-$KLRG1$^-$CD62L$^-$, and cluster #8: CD44$^+$Ki-67$^+$ Ly6C$^+$EOMES$^+$TCF1$^+$CXCR3$^+$PD-1$^-$KLRG1$^-$CD62L$^-$ (Fig. 4b–e). The single-cell trajectory algorithm Wanderlust[30] predicted that the developmental path progressed from the CD62L$^+$CD44$^-$TCF1$^+$ subset to transitional subsets harboring markers that were upregulated and retained (CD44, Ki-67, and Ly6C) or are eventually downregulated (EOMES, CXCR3, CD43$^{1B11}$) (Fig. 4a, f, g). The initially expressed CD62L and TCF1 were downregulated whereas KLRG1 expression was acquired at a late-stage indicating a progressive effector-memory T (T$_{EM}$) cell differentiation program after booster vaccination (Fig. 4f, g).

**Vaccine-specific conditions drive the differentiation of elicited CD8$^+$ T cells**. To examine whether vaccine-specific conditions or the epitope itself influence the differentiation of the elicited CD8$^+$ T cells, we vaccinated mice with SLP vaccines containing the epitope GP$_{34-41}$ (AVYNFATC) derived from lymphocytic choriomeningitis virus (LCMV) in a similar manner (Supplementary Fig. 2a). As observed with the Spike$_{539-546}$-SLP immunization, each subsequent vaccination with GP$_{34-41}$-SLP increased the magnitude of the vaccine-specific CD8$^+$ T cell response, and this response remained higher after contraction (Supplementary Fig. 2b–d). Moreover, also the GP$_{34-41}$-specific CD8$^+$ T cells in the spleen, liver, and lungs where significantly increased upon each vaccination. In the spleen this was caused by an increment of the circulating GP$_{34-41}$-specific CD8$^+$ T cells, in the liver by the CD69$^+$ GP$_{34-41}$-specific CD8$^+$ T$_{RM}$ cells and in the lungs by an increase in both the circulating as well as the tissue-resident T cells (Supplementary Fig. 2e). In addition, these increments were also reflected by increased polyfunctional cytokine producing GP$_{34-41}$-specific CD8$^+$ T cells in the spleen, liver, and lungs (Supplementary Fig. 2f).

Analysis of the differentiation status of the GP$_{34-41}$-specific CD8$^+$ T cells in the blood circulation using tSNE algorithms also showed differentiation into an effector-memory-like state (CD62L$^-$KLRG1$^+$Ly6C$^+$CX3CR1$^+$) after the 2$^{nd}$ vaccination, which was further manifested after the 3$^{rd}$ vaccination (Supplementary Fig. 2g–i). In addition, high-dimensional clustering analysis of the splenic GP$_{34-41}$-specific CD8$^+$ T cells after booster

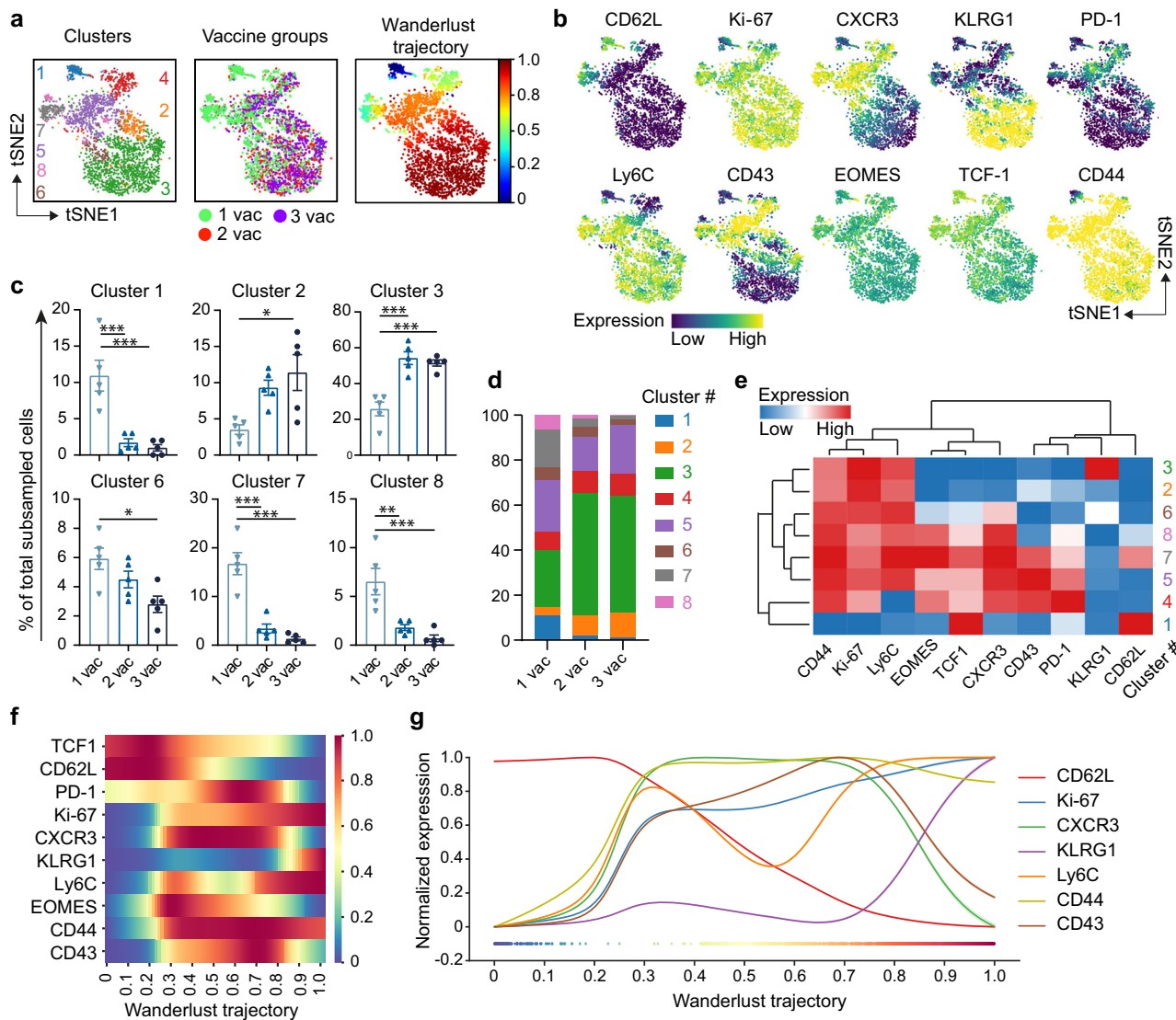

**Fig. 4 Progressive differentiation of vaccine-specific CD8$^+$ T cells after booster vaccination.** C57BL/6 mice were vaccinated subcutaneously on day 0 (1$^{st}$ vaccination), day 14 (2$^{nd}$ vaccination), and day 28 (3$^{rd}$ vaccination) with the Spike$_{539-546}$-SLP vaccine adjuvanted with CpG. **a** Following tSNE analysis downsampling, FlowSOM consensus metaclustering with 8 clusters as well as Wanderlust trajectory analysis was performed on 200 live CD3$^+$CD8$^+$CD4$^-$CD19$^-$ Spike$_{539-546}$ tetramer$^+$ cells harvested from the spleen at day 69 after SLP vaccination. Overlay of the 8 FlowSOM clusters (left), vaccination status (middle) and Wanderlust trajectory (right) (color indication: blue, beginning/early; red, end/late) on the tSNE-map. **b** Expression intensity of cell surface markers (color indication: blue, low expression; yellow, high expression). **c** Significant clusters were selected and are shown in bar graphs. Data represented as mean ± SEM ($n = 5$ per group). Symbols represent individual mice. *$P = 0.011–0.015$, **$P = 0.0043$, ***$P = 0.001–0.009$, ****$P = <0.0001$. **d** Stacked bar graph of all 8 FlowSOM clusters per vaccination group, colors match (**a**). **e** Hierarchically clustered heatmap of phenotypes of the clusters shown in (**a**)—the indicated marker expression is shown per cluster as z-Score of median signal intensity per channel; blue, low expression; red, high expression. **f** Each of the markers used for embedding in figure (**a–c**) are displayed according to Wanderlust trajectory progression. **g** Selected markers are displayed according to Wanderlust trajectory progression. One-way ANOVA with Tukey's multiple comparison test for (**c**). Source data are provided as a Source Data file.

vaccination showed a similar pattern: a large KLRG1$^+$Ly6C$^+$Ki-67$^+$ cluster profoundly present in the 2$^{nd}$ and 3$^{rd}$ vaccination group and less differentiated clusters (CD62L$^+$TCF1$^+$KLRG1$^-$) that were relatively more abundant after the 1$^{st}$ vaccination (Supplementary Fig. 3a–d).

Further insight into the phenotype of the vaccine-specific CD8$^+$ T cells upon sequential vaccination, using single-cell high dimensional mass cytometry with a panel of antibodies targeting markers of cellular activation and differentiation[31] (Supplementary Fig. 4a and Supplementary Table S3), confirmed the disparate phenotype after the 1$^{st}$ vaccination compared to the

2$^{nd}$ and 3$^{rd}$ vaccination (Supplementary Fig. 4b). Principal component analysis (PCA) based on the cluster frequencies of the GP$_{34-41}$-specific memory CD8$^+$ T cells in the spleen, liver and lungs confirmed the clear distinction between clusters present in one-time vaccinated *versus* clusters in two or three times vaccinated mice (Supplementary Fig. 4c). In addition, dual tSNE analysis on all samples of the GP$_{34-41}$-specific CD8$^+$ T cells, visualizing the segregation based on vaccination-associated patterns, corroborated that the 2$^{nd}$ and 3$^{rd}$ vaccination-specific phenotypes had considerable overlaps (Supplementary Fig. 4d). A low Jensen-Shannon (JS) divergence distance was calculated when

comparing antigen-specific CD8$^+$ T cells induced upon 2$^{nd}$ and 3$^{rd}$ vaccinations, indicating a high similarity between cells induced by these vaccinations, whereas a higher JS distance was apparent when comparing 1$^{st}$ to 2$^{nd}$ vaccinations or 1$^{st}$ to 3$^{rd}$ vaccinations (Supplementary Fig. 4e). Visualization and quantification of the particular marker expression on the GP$_{34-41}$-specific CD8$^+$ T cell population indicated that in the spleen and lungs, the percentage CD69$^-$CD62L$^-$ T$_{EM}$-like cells expressing high levels of KLRG1, Ly6C, and CX3CR1, and moderate expression of NKG2A, Sca-1 and the ectonucleotidase CD39 associated to the 2$^{nd}$ and 3$^{rd}$ vaccination, while the CD69$^-$CD62L$^+$ subsets related to single-dose vaccination (Supplementary Fig. 4f±h). The phenotype of CD69$^+$ GP$_{34-41}$-specific CD8$^+$ T$_{RM}$ cells in the liver was slightly affected upon the 3$^{rd}$ vaccination, and resulted in reduction of CD11c$^+$CD122$^+$PD-1$^-$ T$_{RM}$ cells and an increase in CD11c$^+$CD122$^-$PD-1$^+$ T$_{RM}$ cells (Supplementary Fig. 4h). Based on the induction of a highly similar phenotype by the Spike$_{539-546}$-SLP and GP$_{34-41}$-SLP vaccines, we conclude that the differentiation of the vaccine-elicited CD8$^+$ T cells depends on (repeated) vaccine-specific conditions rather than the antigens itself.

**A third vaccination triggers the remigration of T$_{RM}$ cells into the circulation**. To fate map the T$_{RM}$ cell progeny upon booster vaccination, we used a reporter system based on the T$_{RM}$-restricted transcription factor Hobit to evaluate the development of the T$_{RM}$ cells[32]. Here, Hobit lineage tracer (LT) mice, which were generated by crossing CD45.1 Hobit reporter OT-I mice with ROSA26-eYFP mice, were used. CD8$^+$ T cells in these mice recognize the epitope SIINFEKL (OVA$_{257-264}$) and report active Hobit expression (by tdTomato expression) and previous Hobit expression (by YFP expression), which allows the detection of ex-Hobit$^+$ cells (ex-T$_{RM}$; YFP$^+$tdTomato$^-$). First, we tested vaccination with OVA$_{257-264}$-SLP in a prime-boost-boost setting, and this resulted in vaccine-specific CD8$^+$ T cell responses that are comparable to other SLPs with respect to the kinetics and magnitude of the response (Fig. 5a) as well as the induction of the KLRG1$^+$CD62L$^-$ phenotype after the 1$^{st}$ booster vaccination (Fig. 5b), which further corroborates that vaccine-specific conditions drive the differentiation of elicited CD8$^+$ T cells.

Next, we adoptively transferred the OT-I CD8$^+$ T cells of the Hobit LT mice into naïve wild-type mice that received subsequent OVA$_{257-264}$-SLP vaccinations in a prime-boost-boost setting. As expected, an increase of CD8$^+$ T$_{RM}$ cells (CD69$^+$ YFP$^+$tdTomato$^+$) was observed in the liver after the 2$^{nd}$ and 3$^{rd}$ vaccination, which were all co-expressing CD38 (Fig. 5c, d). Remarkably, in particular, the 3$^{rd}$ vaccination induced the emergence of ex-T$_{RM}$ cells in the liver, spleen, and blood circulation, and the formation of these cells was characterized by the KLRG1$^+$CD44$^+$ phenotype (Fig. 5d–g). Thus, ex-T$_{RM}$ cells with a KLRG1$^+$ T$_{EM}$ phenotype are induced upon sequential SLP vaccination and are systemically present in the circulation.

## Discussion

Here, we have investigated the capacity of single B cell and T cell epitope-containing peptide vaccines to elicit protection against SARS-CoV-2 infection in the K18-hACE2 transgenic mouse model, and found that only a third vaccination with a long peptide harboring a single T cell epitope provided full protection. To our knowledge, this is the first study to show that vaccine-elicited CD8$^+$ T cells without the aid of virus-specific CD4$^+$ helper T cells or neutralizing antibodies can protect against SARS-CoV-2, albeit provided in a booster setting requiring at least 2 boosters. These results may be of interest for the current discussion regarding a third vaccination, as the current vaccines elicit virus-specific T cells[4]. In addition, these results may also

guide the development of T cell focused vaccines, which could be of utmost importance for susceptible patient groups such as transplantation and leukemia patients, and patients with auto-immune diseases treated with anti-CD20 antibodies such as rituximab. These patients have impaired antibody responses, and therefore may depend on robust T cell-eliciting vaccines for protection.

The DNA vaccine platform we used encoding Spike protein was highly efficient in generating neutralizing antibodies and resulted in protection against SARS-CoV-2 infection. Although, our results indicated that antibody responses to single linear B cell epitopes are inferior for the induction of neutralizing antibodies, we cannot exclude the possibility that single linear B cell epitopes exist that can elicit neutralizing antibodies. In addition, in settings in which antibodies mediate protection by other mechanisms, such as antibody-dependent cellular cytotoxicity (ADCC), antibody-dependent cellular phagocytosis (ADCP), and complement-dependent cytotoxicity (CDC)[33], the B cell-SLP platform may be valuable. In this respect, the addition of CpG and IFA as adjuvants and the insertion of a CD4$^+$ T helper cell epitope to the linear B cell epitope vaccine was indispensable to elicit antibody responses.

In-depth studies using the T cell epitope SLP vaccines indicated that a third vaccination resulted in superior generation of CD8$^+$ T$_{EM}$ cells, as exemplified by the expression of KLRG1, Ly6C, and the chemokine receptors CXCR3 and CX3CR1, in the circulation and also of CD8$^+$ T$_{RM}$ cells in liver and lungs. The CD8$^+$ T cells elicited upon a third dose vaccination are functionally and phenotypically modulated as evidenced by activation-associated cell-surface markers and their polyfunctional cytokine production. The latter is in line with a recent study in humans, which indicated that a third vaccination in kidney transplant recipients leads to increased circulating polyfunctional CD4$^+$ T cells[34]. It would be of interest to decipher whether a third dose of the mRNA vaccine in healthy individuals, which is more effective against severe COVID-19-related outcomes compared to two doses[35], also associates to increased T cell immunity. It is also of interest to examine whether differential time-intervals between the prime and booster vaccinations have impact on the magnitude and phenotype of the vaccine-elicited T cells[36,37]. A recent study indicated only differences for the induction of IL-2-expressing CD4$^+$ T cell responses by mRNA vaccines when comparing intervals of 3–6 weeks to 8–16 weeks between the 1$^{st}$ and 2$^{nd}$ dose but shorter intervals were not compared[37]. Strikingly, the phenotype and cytokine polyfunctionality was highly similar between the different SLP vaccines, indicating that vaccine-specific conditions are dominant in shaping the T cell phenotype rather than the epitope used. This is in line with other studies showing that the environmental cues during T cell activation dictate the differentiation of these cells[31,38,39].

The increase of T$_{EM}$ and T$_{RM}$ cells in the lungs after the third vaccination as reported here may be critical as the lungs are the primary entry point for SARS-CoV-2. Nevertheless, the circulating vaccine-induced T cells in the spleen and blood may also contribute to protection as well because these cells can rapidly migrate into the infected lung tissue and exert effector function. Moreover, also the efficient formation of the T$_{RM}$ cells in the liver, which was mainly observed after the third vaccination, may contribute to protection as these cells have superior potential to differentiate into ex-T$_{RM}$ cells[40].

In conclusion, a third vaccination with a synthetic vaccine containing a single CD8$^+$ T cell epitope results in protection against SARS-CoV-2 in the absence of neutralizing antibodies. This protection can be explained by an improved quantitative and qualitative CD8$^+$ T cell response after the third vaccination that is highlighted by higher numbers of virus-specific T$_{EM}$ and T$_{RM}$ cells with polyfunctional cytokine capacity.

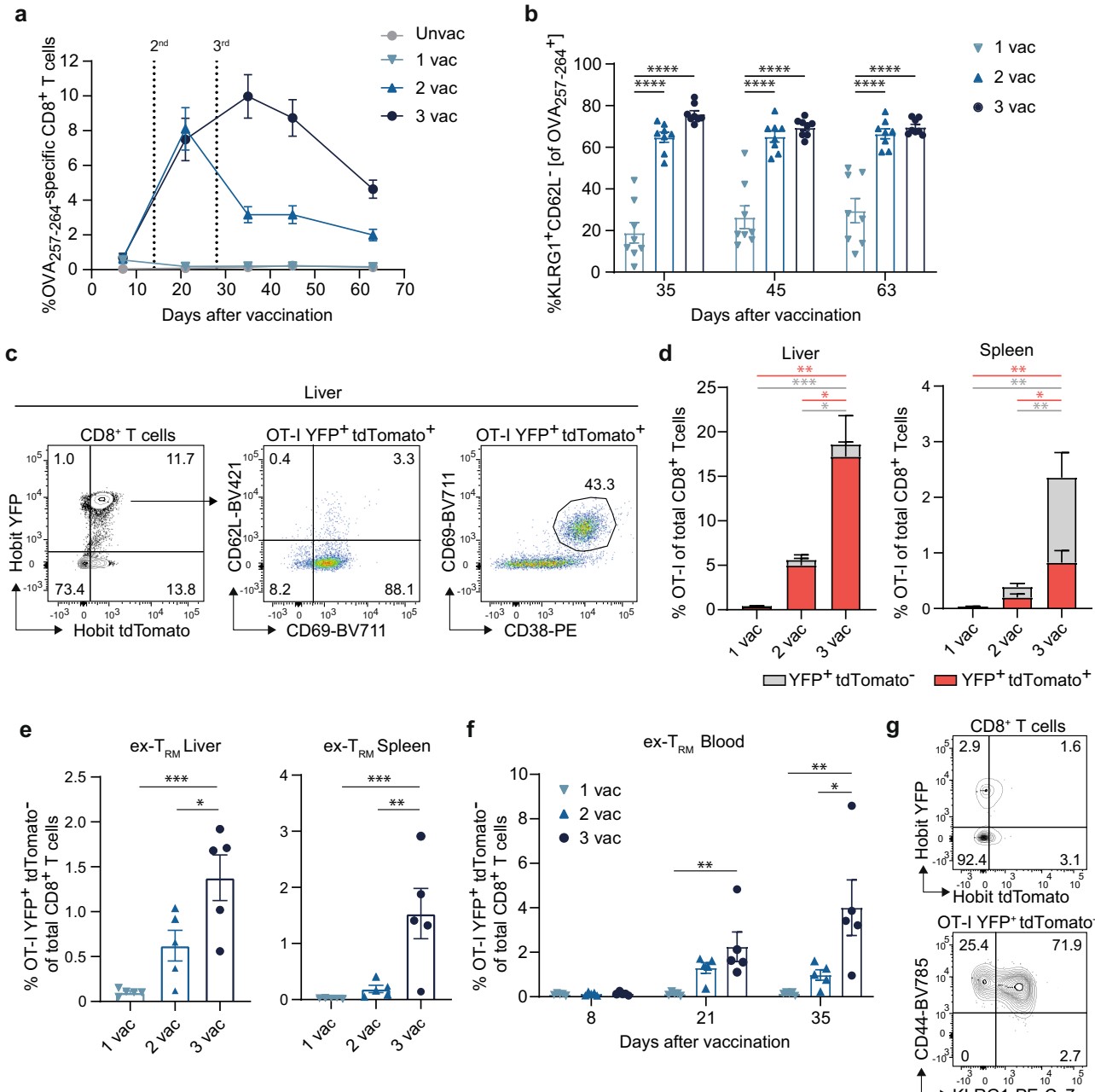

**Fig. 5 A third vaccination triggers the remigration of T$_{RM}$ cells into the circulation. a** C57BL/6 mice were vaccinated with the T cell OVA$_{257-264}$-SLP vaccine adjuvanted with CpG in a prime-boost-boost regimen with 2 week intervals. Shown are the OVA$_{257-264}$-specific CD8$^+$ T cell kinetics in blood at indicated days after vaccination. Data is represented as mean ± SEM ($n = 8$ per group). **b** Frequencies of KLRG1$^+$CD62L$^-$ OVA$_{257-264}$-specific CD8$^+$ T cells in blood in time. Data represented as mean ± SEM ($n = 8$ per group). Symbols represent individual mice. ****$P = < 0.0001$. 1 × 10$^4$ OT-I Hobit LT CD8$^+$ T cells were adoptively transferred into C57BL/6 mice. The day after, mice received the first OVA$_{257-264}$-SLP vaccination, followed by 2 boosters. After 50 days spleen and liver were analyzed by flow cytometry. **c** Flow cytometry plots showing the CD62L, CD69, and CD38 expression of liver OT-I YFP$^+$ tdTomato$^+$ CD8$^+$ T cells. **d** Percentage of OT-I YFP and/or tdTomato positive CD8$^+$ T cells in liver and spleen. Data is represented as mean + SEM ($n = 5$ per group). *$P = 0.0144-0.0267$, **$P = 0.0025-0.0089$, ***$P = 0.0007$. **e** Percentage of OT-I YFP$^+$tdTomato$^-$ (ex-T$_{RM}$) of total CD8$^+$ T cells in liver and spleen. Data represented as mean ± SEM ($n = 5$ per group). Symbols represent individual mice. *$P = 0.0267$, **$P = 0.0040-0.0089$, ***$P = 0.00070082$. **f** Percentage of OT-I Hobit YFP$^+$tdTomato$^-$ (ex-T$_{RM}$) cells of total CD8$^+$ T cells in blood. Data represented as mean ± SEM ($n = 5$ per group). Symbols represent individual mice. *$P = 0.0327$, **$P = 0.0076-0.0082$. **g** Phenotype of OT-I YFP$^+$tdTomato$^-$ (ex-T$_{RM}$) CD8$^+$ T cells in the spleen. One-way ANOVA with Tukey's multiple comparison test for (**a**, **d–f**). Source data are provided as a Source Data file.

## Methods

**Mice**. Wild-type C57BL/6 mice were obtained from Charles River Laboratories, Jackson Laboratory, or Janvier Labs. The K18-hACE2 transgenic mice, expressing the human ACE2 receptor (hACE2) under control of the cytokeratin 18 (K18) promoter[41], were obtained from the Jackson Laboratory (B6.Cg-Tg(K18-ACE2) 2Prlmn/J), and bred in-house. The Hobit reporter OT-I × ROSA26-eYFP mice were previously described[32]. At the start of the experiments, male and female mice were 6-8 weeks old. Animals were housed in individually ventilated cages under specific-pathogen-free conditions at the animal facility at the Leiden University Medical Center (LUMC) at 20–22 °C, a humidity of 45–65% RV, and a light cycle of 6:30 h–7:00 h sunrise, 07:00 h–18:00 h daytime and 18:00 h–18:30 h sunset. All animal experiments were approved by the national Central Animal Testing

Committee (permit number AVD1160020186804), the Animal Tests Committee, and the Animal Welfare Body of the LUMC, and performed according to the recommendations and guidelines set by LUMC and by the Dutch Experiments on Animals Act.

**Peptide and DNA-based vaccination**. SARS-CoV-2 linear B cell epitopes, Spike$_{539-546}$-SLP, GP$_{34-41}$-SLP, and OVA$_{257-264}$-SLP (aa sequences in Supplementary Table 1) were produced at the peptide facility of the LUMC. The purity of the synthesized peptide (75–90%) was determined by HPLC and the molecular weight by mass spectrometry. For the linear B cell epitope studies, mice were vaccinated subcutaneously (s.c.) at the flank with 150 μg SLP + 20 μg CpG (ODN 1826, InvivoGen) dissolved in PBS and emulsified at a 1:1 ratio with Incomplete Freunds Adjuvant (Sigma Aldrich). For the CD8+ T cell SLPs, mice were vaccinated s.c. at the tail base with 100 μg of the Spike$_{539-546}$, GP$_{34-41}$ or OVA$_{257-264}$-SLP + 20 μg CpG (ODN 1826, InvivoGen) dissolved in PBS. Booster vaccinations were provided with 2 weeks interval. For DNA-based vaccination, a codon-optimized, synthetic gene encoding the Spike protein of SARS-CoV-2 was generated. Plasmids were propagated in *E. coli* cultures and purified using Nucleobond Xtra maxi EF columns (Macherey-Nagel) according to the manufacturer's instructions. For vaccination, plasmids were column-purified twice, each time using a fresh column. Mice were intradermally vaccinated at the tail base with a total volume of 30 μL, containing 50 μg DNA in Tris-buffered saline (1 mM Tris, 0.9% NaCl). Booster vaccinations were provided with 3 weeks interval.

**SARS-CoV-2 infection**. Clinical isolate SARS-CoV-2/human/NLD/Leiden-0008/2020 (here named SARS-CoV-2) was used for the SARS-CoV-2 infection of mice. The next-generation sequencing data of this virus isolate is available under GenBank accession number MT705206.1 and shows one mutation in the Leiden-0008 virus Spike protein compared to the Wuhan spike protein resulting in Asp>Gly substitution at position 614 (D614G). In addition, several non-silent (C12846U and C18928U) and silent mutations (C241U, C3037U, and C1448U) in other genes were found. Isolate Leiden-0008 was propagated and titrated in Vero-E6 cells (ATCC CRL-1586). K18-hACE2 transgenic mice were anaesthetized with isoflurane gas and intranasally infected with $5 \times 10^3$ plaque forming units (PFU) of SARS-CoV-2 in a total volume of 50 μl DMEM. Mouse weight and clinical discomfort were monitored daily. Euthanasia criteria were weight loss of >20 percent of body weight compared to the pre-study weight and a moribund state. All experiments with SARS-CoV-2 were performed in the Biosafety Level 3 (BSL3) Laboratories at the LUMC.

**Antigen-binding ELISA**. ELISAs were performed to determine antibody titers in sera. Nunc ELISA plates were coated with 1 μg/ml Spike S1 + S2ECD-His recombinant protein (SinoBiologicals, 40589-V08B1) in ELISA coating buffer (Biolegend) overnight at 4 °C. Plates were washed five times and blocked with 1% bovine serum albumin (BSA) in PBS with 0.05% Tween-20 for 1 h at room temperature. Plates were washed and incubated with serial dilutions of mouse sera and incubated for 1 h at room temperature. Plates were again washed and then incubated with 1:4000 dilution of horse radish peroxidase (HRP) conjugated anti-mouse IgG secondary antibody (SouthernBiotech, cat. 1030-05) and incubated for 1 h at RT. To develop the plates, 50 μL of TMB 3,3=,5,5=tetramethylbenzidine) (Sigma-Aldrich) was added to each well and incubated for 5 min at room temperature. The reaction was stopped by the addition of 50 μL 1 M H$_2$SO$_4$, and within 5 min the plates were measured with a microplate reader (model 680; Bio-Rad) at 450 nm.

**Virus neutralization assay**. Serum was heat-inactivated for 0.5 h at 56 °C, two-fold serially diluted, and incubated with 100 PFU/100 μl of SARS-CoV-2 Zagreb isolate (hCoV-19/Croatia/ZG-297-20/2020, GISAID database ID: EPI_ISL_451934) for 1 h at room temperature. Thereupon, $2 \times 10^4$ Vero-E6 (ATCC CRL-1586) cells that were seeded in 96-well plates were inoculated with serum and virus. Following 1 h incubation at 37 °C and 5% CO$_2$, the inoculum was removed, and 1.5% methylcellulose overlay was added. Infected cells were incubated for 3 days at 37 °C and 5% CO$_2$, prior to crystal violet staining and plaque counting.

**Single-cell preparations**. Peripheral blood was collected from the tail vein. Splenocytes were obtained by mincing the tissue through cell strainers. Blood cells and splenocytes were depleted of erythrocytes using ammonium chloride lysis buffer. To remove remaining circulating blood cells from the liver and lungs, mice were perfused with 20 ml PBS containing 2 mM EDTA. Next, liver and lungs were cut into small pieces using surgical knives, and the tissue was resuspended in 3.5 ml or 1 ml, respectively, of IMDM containing 250 U/ml collagenase type 1-A (C2674, Sigma) and 20 μg/ml DNase I (D5025, Sigma). After incubation with collagenase/DNase for 25 min at 37 °C, liver and lung tissue was dissociated into single-cell suspensions using 70 μm cell strainers, and subsequently lymphocytes were isolated using a Percoll (GE Healthcare) gradient.

**Flow cytometry**. Following resuspension in staining buffer (PBS + 1% FCS) the cells were incubated with fluorescently labeled antibodies purchased from BioLegend, BD Biosciences, Invitrogen, or ThermoFisher (eBioscience). All antibodies used are listed in Supplementary Table 2 and in the Nature Research Reporting Summary linked to this article. Zombie Aqua (BioLegend) staining was used to exclude dead cells. To stain for EOMES and TCF1, cells were fixed for 30 min using the Fixation/Permeabilization concentrate (Invitrogen, FOXP3/Transcription factor Staining buffer set), diluted 1:6 in Fixation/Perm diluent (eBioscience). Following 2 washes in Permeabilization Buffer, anti-TCF1 and anti-EOMES antibodies (diluted in Permeabilization Buffer) were added and the cells were incubated for 1 h. Following a washing step in staining buffer the cells were ready for analysis. Glucose uptake was measured by the uptake of glucose analog 2-(N-(7-nitrobenz-2-oxa-1,3-diazol-4-yl)amino)-2-deoxyglucose (2-NBDG, Life Technologies). Cell surface and intracellular cytokine stainings of splenocytes and blood lymphocytes were performed as described[42]. PADRE-specific CD4+ T cells were quantified using an MHC class II tetramer for the epitope (AKFVAAWTLKAA) (NIH Tetramer Core Facility). Spike$_{539-546}$-specific, GP$_{34-41}$-specific or OVA$_{257-264}$-specific CD8+ T cells were quantified using MHC class I tetramers generated in-house. For examination of intracellular cytokine production, single cell suspensions were stimulated with short peptides for 5 h in the presence of brefeldin A or with long peptides for 8 h of which the last 6 h in presence of brefeldin A (Golgiplug; BD Pharmingen). Flow cytometric acquisition was performed on a BD Fortessa flow cytometer (BD Biosciences) with BD FACS DIVA software (version 9) or on the Aurora Cytek spectral analyzer with SpectroFlo acquisition software (version 3), and samples were analyzed using FlowJo software (TreeStar) and OMIQ data analysis software (Gating strategy shown in Supplementary Fig. 5).

**Adoptive T cell transfers**. Spleens of OT-I Hobit reporter × ROSA26-eYFP LT mice were isolated and subsequently, OT-I CD8+ T cells were isolated by negative selection using MicroBeads (130-104-075, Miltenyi Biotec) according to the manufacturer's protocol. Splenic OT-I CD8+ T cells were adoptively transferred via retro-orbital injection into naïve mice.

**Statistical analysis**. Statistical analyses were performed using Cytofast (version 1.6.0) or GraphPad Prism (version 8, La Jolla, CA, Unites States). One-way ANOVA and log-rank (Mantel–Cox) survival test were used for statistical analysis. All *P* values were two-sided, and $P < 0.05$ was considered statistically significant.

**Reporting summary**. Further information on research design is available in the Nature Research Reporting Summary linked to this article.

## Data availability
The authors declare that the data supporting the findings of this study are available in the article, Supplementary Information or Source Data file. Source data are provided with this paper. The different SARS-CoV-2 variants used in this paper can be found in the GISAID database ID: EPI_ISL_451934 (https://www.gisaid.org/) and the GenBank database: accession number MT705206.1. Source data are provided with this paper.

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

## Acknowledgements

This work was funded by Health Holland (LSH-TKI project LSHM20036) and the Dutch Research Council (NWO-TTW, grant number 15830) awarded to R. Arens; The Gisela Thier Fellowship to T.C. van der Sluis, LUMC (BWplus) and the graduate program of the Dutch Research Council to E.T.I. van der Gracht; Helmholtz Association's Initiative and Networking Fund (the Project "Virological and immunological determinants of COVID-19 pathogenesis – lessons to get prepared for future pandemics (KA1-Co-02 "COVIPA")") to L. Cicin-Sain. The authors would like to thank Prof. Dr. Jannie Borst for critically reading the manuscript, and the Experimental Animal and Flow Cytometry facilities for their support.

## Author contributions

I.N.P., T.C.v.d.S., E.T.I.v.d.G., D.M.B.V., S.v.D., G.B., F.B., J.R., R.N., E.B.N., N.M., D.B., and M.G.M.C. performed the experiments and analyzed the data; I.N.P., E.T.I.v.d.G., and R.A. designed the experiments; Y.K. and L.C.-S. performed the antibody neutralization studies; S.K.M., P.J.B., and M.K. contributed to the SARS-CoV-2 challenge studies; T.A. performed bioinformatics analyses; K.L.M.C.F., K.P.J.M.v.G., and G.C.M.Z. provided essential reagents; C.J.M.M., G.C.M.Z., J.W.D., and F.O. contributed to study design for the vaccination studies; I.N.P., T.C.v.d.S., E.T.I.v.d.G., and R.A. interpreted the data, prepared the figures and wrote the manuscript; R.A. conceived and supervised the study.

## Competing interests

C.J.M.M. is Chief Scientific Officer of ISA Pharmaceuticals, a biotech company developing novel therapeutic vaccines against cancer and virus infections. G.C.M.Z. is employee of Immunetune BV, a company developing DNA vaccines against cancer and coronaviruses. All other authors declare no competing interests exist.
