## [Peer Review File · Nature Communications]

A third vaccination with a single T cell epitope confers protection in a murine model of SARS-CoV-2 infectionREVIEWER COMMENTS

Reviewer #1 (Remarks to the Author):

An interesting report that assesses the potential of a 3 course DNA vaccine with a single epitope to protect transgenic mice from SARS-CoV-2 challenge.

The work is in an important area as T cell immunity to SARS-CoV-2 has been difficult to study and is highly important in relation to current understanding of disease protection.

Points

-I am not sure whether the initial work with the B cell epitope SLPs adds anything to the story. None of these responses were neutralising. Use of the whole spike protein (as in current human vaccines) was protective. This just seems to suggest that these are poor SLPs that generate weak spike-specific antibodies.

This can lead to potentially misleading statements as to the potential importance of antibodies. As in the Summary "A third dose with a single T cell epitope-vaccine promotes ... while single B cell epitope-eliciting vaccines are unable to provide protection.

It is perhaps a little surprising that there was no statistical improvement in survival with 2 doses. And then all mice were saved after 3 doses. It may all depend on statistics and number of mice challenged. The mice clearly had a severe disease with 20% weight loss in all cases. I am loathe to suggest further work, with the associated ethical considerations, but the interpretation does need to be tempered somewhat. Number of animals should be stated in methods or figure legends.

The vaccine regimen is very different from current human studies of course - with short time interval and DNA delivery. This should be discussed.

Please put some numerical values in the Results section - eg what percentage increase in spleen after 3 vaccines etc.

The great majority of the subsequent work (Figure 3C onwards) relates to the LCMV epitope. Hopefully this will be correlate with results for other CD8+ epitopes (such as SARS-CoV-2) using this platform but that is far from assured and will relate to factors such as TCR affinity, peptide presentation. This should be discussed.

Reviewer #2 (Remarks to the Author):

The manuscript by Pardieck et investigates if vaccination with a single T cell epitope derived from the spike protein of SARS-CoV-2 is able protect mice from a SARS-CoV-2 challenge following a triple vaccination regime. The authors detail experiments to determine if vaccination using linear B cell epitopes derived from spike to induce antibody responses are protective. Initial observations show that in themselves these do not generate antibody, that providing CD4 T cell help induces some antibody and supplementing this with adjuvants (CpG and IFA) increases CD4 help and induces greater antibody response. However, the antibodies induced can not neutralize the virus and are are not protective in the murine challenge model, in contrast to a whole spike DNA vaccine.

The authors then examine a triple vaccination regime using a single CD8 T cell peptide epitope derived from spike and show that this can induce complete protection in the challenge model. Further investigation of the epitope specific T cells generated show them to increase in frequency following each boost and to generate a resident memory population in tissues. Further experiments demonstrate that triple vaccination with an unrelated LCMV T cell peptide also generated similar T cell responses. The LCMV model was then used to perform in depth deep phenotyping of the T cells using mass cytometry. Finally, the model was changed to OVA CD8 T cell epitope (which also induced a similar T cell response with generation of Trm cells after 3 vaccinations) and a murine system that allows fate mapping using previous and active Hobbit transcription factor expression to show that previous specific Trm cells could then become Tem cells.

I am unsure what the value of the results from experiments with the linear B cell epitope gives to this particular manuscript. It is clear that a DNA vaccine encoding spike induces neutralizing antibodies as well as both CD4 and CD8 T cells, this is also well established for the RNA based COVID19 vaccines now extensively used in humans. The authors do not making a compelling argument in the paper for inclusion of this data. The rest of the paper then focuses on the generation of CD8 T cell responses, given the data from the results in Fig 1 it is unlikely this small epitope will induce antibodies and this is indeed shown to be the case.

The argument that it is potentially important to investigate if CD8 T cell responses alone could protect against SARS-CoV-2 infection in individuals that might not be able to make affective antibody responses is valid and of interest. The results do indeed show, in this particular model, that adjuvanted multiple boosts of peptide do induce specific T cells and these correlate with protection against SARS-CoV-2 challenge (Fig 2 and Fig 3 A and B). The detailed characterization of T cells is however only done on LCMV specific cells and not SARS-CoV-2 specific cells and the fate mapping results has to shift antigen again to OVA. It is difficult to understand why the detailed mass cytometry was not also performed on the spike 539-546 specific cells and as such the data from the LCMV specific cells is inferred to be true for the spike specific cells, this does detract from the paper.

The DNA vaccine encoding whole spike was used as a control and was also shown to generated CD8+ T cells, given that this would include the spike 539-546 peptide epitope was the spike 539-546 specific tetramer used to detect and measure these cells following DNA vaccination? If these tetramer cells are generated, do they also show increased frequency on boosting? Do they have similar phenotypes with regard to Trm cells seen with the spike single epitope vaccinations? These would seem to be import and tractable questions which would significantly increase the interest in the study and its potential applicability to use in humans given the groups speculation about of human vaccine 2 dose and booster regimes.

Minor comments

Fig 1

Panel A details 4 preparations E, E+P,P-E and E-P the figure legend only describes E, E-P and E+P. In addition the figure shows that E-P induces the most superior antibody response but should this not be E+P which is the preparation with both adjuvants in?

Panel B legend text does not state that this is the SLP and PADRE in isolation, with individual each adjuvant individually and finally the mixture.

Panel H it is unclear what this panel adds to the figure as G already shows that most animals die irrespective of the vaccine. The data from the DNA vaccine control would be better here rather than a supplementary figure as this is now an essential control to show that the mice can be protect from the challenge.

Response to referees; Point-by-point reply.

We would like to thank the reviewers for their constructive comments and suggestions to improve the manuscript.

Reviewer #1 (Remarks to the Author):

An interesting report that assesses the potential of a 3 course DNA vaccine with a single epitope to protect transgenic mice from SARS-CoV-2 challenge.

The work is in an important area as T cell immunity to SARS-CoV-2 has been difficult to study and is highly important in relation to current understanding of disease protection.

Points

-I am not sure whether the initial work with the B cell epitope SLPs adds anything to the story. None of these responses were neutralising. Use of the whole spike protein (as in current human vaccines) was protective. This just seems to suggest that these are poor SLPs that generate weak spike-specific antibodies. This can lead to potentially misleading statements as to the potential importance of antibodies. As in the Summary "A third dose with a single T cell epitope-vaccine promotes while single B cell epitope-eliciting vaccines are unable to provide protection.

We fully agree that the presented work should not lead to potentially misleading statements as to the potential importance of antibodies. Our work with the DNA vaccines actually shows that these vaccines elicit neutralizing antibodies leading to protection. Our work with the SLPs indicated that linear B cell epitopes in Spike elicit antibodies that are not neutralizing and are therefore not effective. We have re-phrased the text including the title of the paragraph: "Vaccine-elicited antibodies against conformational proteins but not linear B cell epitopes are neutralizing and effective against SARS-CoV-2 infection".

-It is perhaps a little surprising that there was no statistical improvement in survival with 2 doses. And then all mice were saved after 3 doses. It may all depend on statistics and number of mice challenged. The mice clearly had a severe disease with 20% weight loss in all cases. I am loathe to suggest further work, with the associated ethical considerations, but the interpretation does need to be tempered somewhat. Number of animals should be stated in methods or figure legends.

As suggested we have tempered the interpretations and added all specifics of the experiments (including number of animals in methods and figure legends). In addition, we have submitted all data in the source data file.

-The vaccine regimen is very different from current human studies of course - with short time interval and DNA delivery. This should be discussed.

This is a very interesting note from the reviewer and we have commented on this in the discussion section (page 13, line 5-10): "It is also of interest to examine whether the differential time-intervals between the prime and booster vaccinations impact on the magnitude and phenotype of the vaccine-elicited T cells. A recent study indicated only differences for the induction of IL-2 expressing CD4⁺ T

cell by mRNA vaccines when comparing intervals of 3-6 weeks to 8-16 weeks between the first and second dose but shorter intervals were not compared”.

-Please put some numerical values in the Results section - eg what percentage increase in spleen after 3 vaccines etc.

We have added numerical values in the results sections and have chosen to indicate the fold increase after booster vaccination.

-The great majority of the subsequent work (Figure 3C onwards) relates to the LCMV epitope. Hopefully this will be correlate with results for other CD8+ epitopes (such as SARS-CoV-2) using this platform but that is far from assured and will relate to factors such as TCR affinity, peptide presentation. This should be discussed.

As suggested by the editor and the reviewers we have put focus on the in-depth phenotypic analysis of the SARS-CoV-2 specific T cells in the revised manuscript. We have included novel data on the phenotype of the SARS-CoV-2 specific T cells (new Figure 3 and 4) and compared this to the LCMV epitopes (all data with the LCMV epitope is now placed in the supplementary figures). We have included a paragraph comparing the different platforms (page 13, lines 10-13): “ Strikingly, the phenotype and cytokine polyfunctionality was highly similar between the different SLP vaccines, indicating that the vaccine-specific conditions are dominant in shaping the T cell phenotype rather than the epitope used. This is in line with other studies showing that the environmental cues during T cell activation dictates the differentiation of these cells”.

Given the availability of the mouse strains, the data regarding the HOBIT reporter could only be accomplished with the OT-I system. As the OVA epitope provides a highly similar outcome compared to SARS and LCMV epitopes with respect to magnitude and phenotype upon SLP vaccination, we expect the mechanisms regarding the induction of ex-TRM are extrapolatable.

Reviewer #2 (Remarks to the Author):

The manuscript by Pardieck et al investigates if vaccination with a single T cell epitope derived from the spike protein of SARS-CoV-2 is able to protect mice from a SARS-CoV-2 challenge following a triple vaccination regime. The authors detail experiments to determine if vaccination using linear B cell epitopes derived from spike to induce antibody responses are protective. Initial observations show that in themselves these do not generate antibody, that providing CD4 T cell help induces some antibody and supplementing this with adjuvants (CpG and IFA) increases CD4 help and induces greater antibody response. However, the antibodies induced can not neutralize the virus and are not protective in the murine challenge model, in contrast to a whole spike DNA vaccine.

The authors then examine a triple vaccination regime using a single CD8 T cell peptide epitope derived from spike and show that this can induce complete protection in the challenge model. Further

investigation of the epitope specific T cells generated show them to increase in frequency following each boost and to generate a resident memory population in tissues. Further experiments demonstrate that triple vaccination with an unrelated LCMV T cell peptide also generated similar T cell responses. The LCMV model was then used to perform in depth deep phenotyping of the T cells using mass cytometry. Finally, the model was changed to OVA CD8 T cell epitope (which also induced a similar T cell response with generation of Trm cells after 3 vaccinations) and a murine system that allows fate mapping using previous and active Hobbit transcription factor expression to show that previous specific Trm cells could then become Tem cells.

-I am unsure what the value of the results from experiments with the linear B cell epitope gives to this particular manuscript. It is clear that a DNA vaccine encoding spike induces neutralizing antibodies as well as both CD4 and CD8 T cells, this is also well established for the RNA based COVID19 vaccines now extensively used in humans. The authors do not making a compelling argument in the paper for inclusion of this data. The rest of the paper then focuses on the generation of CD8 T cell responses, given the data from the results in Fig 1 it is unlikely this small epitope will induce antibodies and this is indeed shown to be the case.

The concern of the reviewer has also been raised by reviewer 1. In line with the suggestions by the editor, we have re-phrased the results and conclusions regarding the experiments with the linear B cell epitope, and added data of the DNA vaccine including T cell responses. We think that inclusion of the SLP and DNA vaccine data regarding the B cell response is now better framed, and placed in the context of the CD8 T cell epitope vaccinations.

-The argument that it is potentially important to investigate if CD8 T cell responses alone could protect against SARS-CoV-2 infection in individuals that might not be able to make affective antibody responses is valid and of interest. The results do indeed show, in this particular model, that adjuvanted multiple boosts of peptide do induce specific T cells and these correlate with protection against SARS-CoV-2 challenge (Fig 2 and Fig 3 A and B). The detailed characterization of T cells is however only done on LCMV specific cells and not SARS-CoV-2 specific cells and the fate mapping results has to shift antigen again to OVA. It is difficult to understand why the detailed mass cytometry was not also performed on the spike 539-546 specific cells and as such the data from the LCMV specific cells is inferred to be true for the spike specific cells, this does detract from the paper.

We fully agree with the raised concern. Originally we have included the in depth phenotype of the LCMV-specific T cells as the phenotype was highly similar of the SARS-COV-2 specific T cells, which we partly placed supplementary. However, we understand that for conformity and clarity this should be reversed. In the revised manuscript we have included novel in-depth analysis of the SARS-COV-2 specific T cells (new figure 3 and 4) as opposed to the LCMV-specific T cells (now in the supplementary figures). We generated a novel panel of antibodies based on the markers that were relevant according to the mass cytometry panel and additionally added new functional markers including Ki-67 and TCF-1 and used this panel for both SARS-CoV-2 specific T cells as well as LCMV-specific T cells. We analyzed the high-dimensional cytometry data by several means and included the Wanderlust trajectory data to show the phenotypic development upon booster vaccination. The in-depth analysis demonstrated the induction of a specific TEM-like phenotype of the vaccine-elicited T cells after the booster, and this phenotype is observed after both the SARS and LCMV-epitope SLP vaccination. The data regarding the HOBIT reporter could only be accomplished with the OT-I system. As the vaccination with the OVA epitope in the SLP context provides a similar outcome as with the SARS and LCMV epitopes, we expect that the mechanisms regarding the induction of ex-TRM are extrapolatable.

-The DNA vaccine encoding whole spike was used as a control and was also shown to generate CD8+ T cells, given that this would include the spike 539-546 peptide epitope was the spike 539-546 specific tetramer used to detect and measure these cells following DNA vaccination? If these tetramer cells are generated, do they also show increased frequency on boosting? Do they have similar phenotypes with regard to Trm cells seen with the spike single epitope vaccinations? These would seem to be important and tractable questions which would significantly increase the interest in the study and its potential applicability to use in humans given the groups' speculation about of human vaccine 2 dose and booster regimes.

We have also used the tetramers to detect the S539-546-specific T cells following DNA vaccination. In the revised manuscript (Figure 1I) we have added these data showing that the frequencies of the S539-546-specific T increase upon prime/boost DNA vaccination. The S539-546-specific T cell-response remains lower as compared to SLP vaccines and the second DNA vaccine booster appears to be less effective. It seems thus that the DNA vaccine platform is capable to prime T cells but booster vaccines with DNA vaccines work less effectively as SLP vaccines in expanding epitope-specific T cells. Nevertheless, antibody responses are well boosted by the DNA vaccines. Phenotypically, these cells also express KLRG1 albeit in lower frequency as compared to the SLP elicited T cells (new Figure 1J).

Minor comments

-Fig 1

Panel A details 4 preparations E, E+P, P-E and E-P the figure legend only describes E, E-P and E+P. In addition the figure shows that E-P induces the most superior antibody response but should this not be E+P which is the preparation with both adjuvants in? Panel B legend text does not state that this is the SLP and PADRE in isolation, with individual each adjuvant individually and finally the mixture.

We have corrected the figure legend and have indicated now better that E + P is the epitope plus uncoupled PADRE and the PADRE-coupled epitopes are E-P and P-E. For clarity we start this Figure with the role of the adjuvants.

-Panel H it is unclear what this panel adds to the figure as G already shows that most animals die irrespective of the vaccine. The data from the DNA vaccine control would be better here rather than a supplementary figure as this is now an essential control to show that the mice can be protected from the challenge.

We fully agree that the DNA vaccine should be implemented in the main Figure rather than Supplementary. We show the DNA vaccine data and have added the T cell response upon DNA vaccination. For conformity we show the Survival plots as well as the individual weight of the mice.

REVIEWER COMMENTS

Reviewer #1 (Remarks to the Author):

The authors have addressed the Reviewers' comments appropriately.
This is now an important paper for consideration of vaccine policy.

Reviewer #2 (Remarks to the Author):

I would like to thank the authors for addressing my comments and not the new data as well as the re-arrangements made from the supplementary figures.

Response to referees; Point-by-point reply.

Reviewer #1 (Remarks to the Author):

The authors have addressed the Reviewers' comments appropriately.
This is now an important paper for consideration of vaccine policy.

Reviewer #2 (Remarks to the Author):

I would like to thank the authors for addressing my comments and the new data as well as the re-arrangements made from the supplementary figures.

Reply:

We would like to thank the reviewers for their constructive comments and suggestions to improve the manuscript.